# Coping with rejection as a sperm donor: A qualitative study of the personal impact of rejection and new health information

Lina Thirup[1]*, Anne-Bine Skytte[1], Ulrik Schiøler Kesmodel[1,2], Ida Vogel[3,4,5], Guido Pennings[6], Allan Pacey[7], Stina Lou[3,4,8]

1 Cryos International, Aarhus, Denmark, 2 The Fertility Unit, Department of Obstetrics and Gynaecology, Aalborg University Hospital, Aalborg, Denmark, 3 Center for Fetal Diagnostics, Aarhus University, Aarhus, Denmark, 4 Department of Clinical Medicine, Aarhus University, Aarhus, Denmark, 5 Department of Clinical Genetics, Aarhus University Hospital, Aarhus, Denmark, 6 Department of Philosophy and Moral Science, Bioethics Institute Ghent (BIG), Universiteit Gent, Gent, Belgium, 7 Faculty of Biology, Medicine and Health, School of Medical Sciences, University of Manchester, Manchester, United Kingdom, 8 DEFACTUM—Public Health Research, Central Denmark Region, Aarhus, Denmark

* lina-th@hotmail.com

## Abstract

### Purpose

The demand for sperm donation is increasing, yet only a small percentage of applicants are accepted, and little attention has been given to those who are rejected. The application process may reveal new medical or genetic information with potential personal and emotional consequences. The aim of this study was to explore how men experience rejection as sperm donors, including how they cope with the rejection and integrate potential new information into their lives.

### Methods

We conducted qualitative, in-depth interviews with 19 men rejected as sperm donors. Data was analyzed using thematic analysis.

### Results

Some men received new medical information that raised significant concerns, particularly regarding their health, fertility, and family. However, for most it was not the reason for rejection that affected them most, but the rejection itself. They had begun to identify as donors, and being turned down felt like a blow to their sense of self. Over time, participants used different coping strategies: information seeking, actionable reasoning, positive reframing, normalizing, and postponing. Although the experience was personally and emotionally challenging for many, it ultimately did not have a lasting negative impact on their lives.

**Data availability statement:** For this study, participants only consented to external data sharing in anonymized form. Since full transcripts cannot be fully anonymized due to the highly individual context, the transcripts can not be shared in a public repository. However, data can be made available from Cryos International, Aarhus, Denmark (contact via dk@cryosinternational.com) for researchers who meet the criteria for access to confidential data.

**Funding:** This study was funded by Cryos International Sperm & Egg bank. LT was hired by Cryos International Sperm & Egg bank to conduct this study, but the funders had no role in study design, data collection and analysis, decision to publish, or preparation of the manuscript.

**Competing interests:** I have read the journal's policy and the authors of this manuscript have the following competing interests: LT is a former employee of Cryos International Sperm & Egg bank. ABS and USK are employed by Cryos International Sperm & Egg bank. GP and AP are members of the External Scientific Advisory Committee (ESAC) of Cryos. IV and SL have declared that no competing interests exist.

## Conclusions

Sperm donation is not only a medical act but also tied to social identity, with rejection having personal and emotional consequences. For men whose rejection is based on new medical information that causes significant concerns, sperm banks should support rejected donors and help them exit the process positively. Further research is warranted to find out how this can be done.

## Introduction

The need for gamete donation is increasing and the continuous recruitment and retention of men willing to donate sperm is essential for meeting this demand [1]. While declining sperm quality and infertility remain important drivers [1], recent studies show that a substantial and growing proportion of the demand for donor sperm today comes from single women and same-sex couples [2–4]. Men's reasons for becoming sperm donors have been extensively researched, often pointing to financial incentives [5–7], curiosity about sperm quality [8–10], and altruism as main motivating factors [7,9,11–13]. However, little attention has been given to the very large percentage of applicants who are rejected as sperm donors [14,15]. A recent study on sperm donor applicants found that only 4% are accepted, as most applicants drop out or are rejected [16]. When a man applies to become a sperm donor, he must undergo a long application process. Depending on which sperm bank he applies to, it may include initial semen analysis, a health questionnaire, a consent interview, a psychosocial interview, and a clinical examination before being approved as a sperm donor [15]. This application process will, in some cases, result in him gaining new health and medical knowledge about himself, e.g., information about reduced sperm quality, genetic predisposition to disease, and physical illness. Many applicants, however, may not be fully prepared for this new information, which can have significant consequences. It could impact not only themselves but also their relationships with current or future family members, as they may now be confronted with unforeseen health challenges. Understanding the experiences of this large group of men who are not accepted as sperm donors is an important yet largely overlooked area of research. A recent quantitative survey of rejected sperm donor candidates found that the majority were disappointed and surprised, but only few had concerns about their health, fertility, or manhood [15]. However, to more fully understand these feelings and concerns, the aim of this study was to explore men's experience of being rejected as a sperm donor, focusing on how they cope with the rejection and the new information and how they integrate it into their lives.

## Materials and methods

The study was conducted with an explorative study design based on semi-structured qualitative interviews [17,18].

## Study participants

Participants were recruited via Cryos International Sperm & Egg bank (www.cryosinternational.com). Between 26 April 2024 and 16 September 2024, 188 rejected sperm donors were contacted by email by LT, to be invited to participate in the interview study. These 188 donors had been rejected within the previous year (26 April 2023–26 April 2024) and were drawn from a total of 3,031 men who applied to become sperm donors during that period. The inclusion criteria were: 1) Rejected sperm donor, 2) rejected within the last year, 3) rejected in Denmark, 4) consent to being contacted, and 5) available contact information. Sample variation was sought for age, parity, reason for rejection and time since rejection. A total of 24 rejected sperm donors responded to the email, stating they would like to participate in the study. LT contacted them by phone with additional information about the study, answered any questions, and upon consent, an interview appointment was arranged. Out of 24 rejected sperm donors, 5 could not be reached, thus leaving a sample of 19 rejected sperm donors. Participant characteristics are presented in Table 1.

## Data collection

Before all interviews, the participant provided written consent to participate in the interview. All interviews were conducted via video conference with LT as the interviewer, who is experienced in qualitative interviewing. Information about the study was repeated before all interviews and oral consent was obtained at the video conference. It was clearly stated that participation was voluntary, and that consent could be withdrawn at any time during the research process. The interviews were guided by a semi-structured interview guide informed by the interdisciplinary research group and existing international literature. Topics in the interview guide and examples of questions can be found in Table 2. All participants were encouraged

**Table 1. Participant characteristics (N = 19).**

| Mean age in years (range) | 28 (19–42) |
|---|---|
| Partner | |
| Yes | 10 |
| No | 9 |
| Own children | |
| Yes | 4 |
| No | 15 |
| Educational level | |
| No education beyond primary school | 1 |
| Students | 12 |
| Masters | 6 |
| Ethnicity | |
| Danish | 19 |
| Other | 0 |
| Reasons for rejection as a sperm donor | |
| Reduced sperm quality | 6 |
| Physical illness | 5 |
| Genetic predisposition to disease | 4 |
| Mental illness | 2 |
| Lack of information about their family's medical history | 2 |
| Mean time since rejection in months (range) | 8 (5–12) |
| Mean duration of interview in minutes (range) | 33 (17–48) |

**Table 2. Topics and examples of questions in the interview guide.**

| Topic | Example of question |
|---|---|
| Introduction | "Could you start by telling me a little about yourself?"<br>"What were your first thoughts when you received my email about this research project?" |
| Before application | "Could you tell me why you applied to become a sperm donor?"<br>"Could you tell me about the considerations you had before applying?"<br>"Did you talk to anyone about it?" |
| Application | "Could you tell me about your experience with the application process?"<br>"Did you involve anyone in the process?" |
| Rejection | "Why were you rejected?"<br>"How did it feel to receive that information when you hadn't asked for it?"<br>"How did you find the communication?" |
| After the rejection | "Has it led to any thoughts or reflections afterward?"<br>"Did it lead to any specific actions?"<br>"Looking back, is there anything you wish Cryos had provided?"<br>"Do you regret applying?" |

to speak freely and introduce topics that they found relevant. All interviews were digitally recorded, transcribed verbatim, anonymized and lasted on average 33 minutes (range: 17–48 minutes).

## Data analysis

We approached the study inductively, and the material was analysed using reflexive thematic analysis [19,20]. After a thorough reading and open coding of all transcripts, SL and LT developed a list of potential inductive codes, which were then discussed with the co-authors. Following this discussion, the list was refined into a final coding structure. LT systematically coded all material using Nvivo 15 software (QSR International, 2024). Nvivo was used solely as a tool to organize and manage the qualitative data, and did not influence the interpretation of the material. To ensure consistency and rigor, SL and LT subsequently reviewed and summarized the content of all codes, and the relations between the codes and the complete data set were critically discussed by all authors. During this process, the analysis began to center on two particular themes: The meaning and significance of the rejection, and the different strategies the men used to cope with the rejection. We therefore turned to Social Identity Theory [21], and the Stress and Coping Model [22] to provide us with theoretical perspectives and bring the analysis forward. We did not code deductively based on these theories but instead used them in an abductive process to gain a deeper understanding of the material [23]. These theories will be introduced and used in the discussion. The analysis resulted in a total of three main themes, each with two to five subthemes, as illustrated by the thematic map (Fig 1).

## Ethical approval

This study was reviewed by The Central Denmark Region Committees on Health Research Ethics (No. 101/2025). According to Danish legislation, research using questionnaires and interviews that do not involve human biological material (§14(2) of the Committee Act) interview studies are exempt from approval from the Committee on Health Research Ethics (https://en.nvk.dk/how-to-notify/what-to-notify). The study has been reviewed and approved by Cryos International's legal department. Informed consent (written and oral) to participate in the study and to having their data published was obtained from all participating men.

## Results

The analysis resulted in the identification of three themes: "Wanting to do good – for others and myself", "Rejection – I did not see that coming!", and "Life goes on – it is what it is".

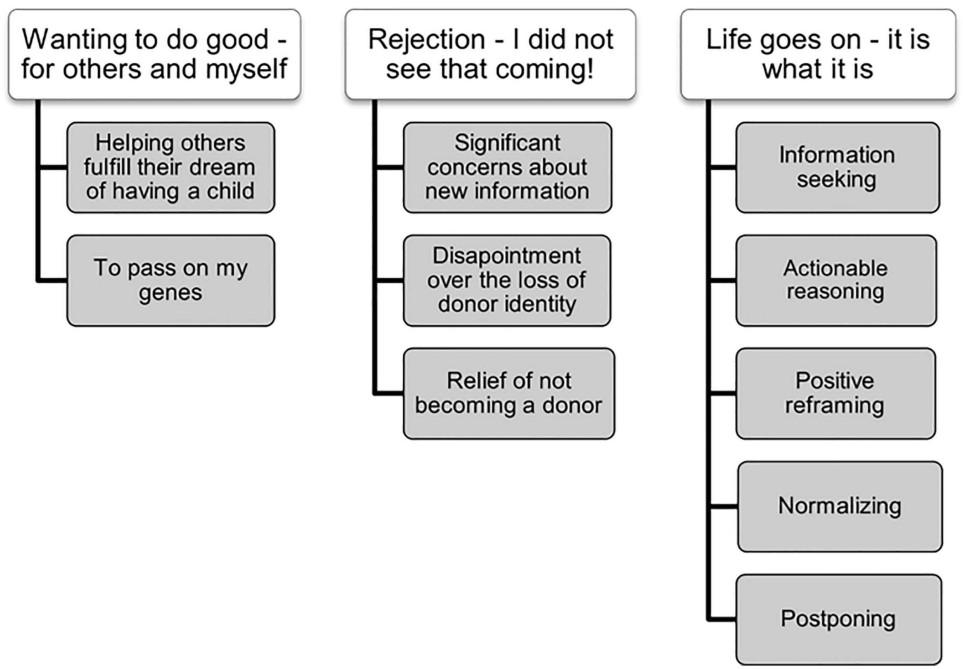

**Fig 1. Themes and subthemes.**

### Wanting to do good – for others and myself

The analysis identified two subthemes: "Helping others fulfill their dream of having a child" and "To pass on my genes".

Nine men had given much thought to becoming sperm donors before applying. For the remaining ten, it had been a more spontaneous decision. However, a key motivation mentioned by most was the altruistic hope of helping others fulfill their dream of having a child. A man in his twenties said: "I think it's a privilege [having children] that everyone should be allowed to have." (IP10). Many of these men's hopes stemmed from personal experience: needing fertility treatment themselves, being donor conceived, or having people in their social network who struggled to conceive. A man in his thirties with a partner and children, said: "We have many friends who struggled a bit with having children. It really affected me, and they were absolutely heartbroken for several years. I felt so sorry for them." (IP9). These men had, in one way or another, up-close experience with the longing for a child.

For nine men, a primary motive for donating sperm was to pass on their genes. Some men described being 'on the verge of giving up' the dream of having children of their own, e.g., due to high age or lack of a partner and thus felt a desire to pass on their genes in other ways. A man in his forties with a partner and no children said: "It was somehow a desire to pass something on. To pass on my genes. I honestly wouldn't mind if someone came forward in 18 years' time wanting to know where they came from, DNA-wise. It isn't frightening. It was almost like, well, 'enticing' is too strong a word, but it would be kind of nice in a way if someone wanted that." (IP4). These men were motivated by the prospect of 'someone out there' who shared a genetic connection with them and maybe would find them interesting. Others did not desire children of their own but still wanted to pass on their genes. A man in his thirties with a partner, said: "We're not interested in being parents in that way, but I still think it could be fun to see what would come of it in some way. Personally, I think it would be interesting if, at some point later on, I had a biological heir who was searching for me or something like that." (IP19). This man did not desire 'own children' but was curious about the potential 'result' of his genes and the possibility of some form of future contact. Another man, in his twenties and a self-proclaimed 'redhead', wanted to pass on his

genes to ensure 'the redhead genes' would not become extinct: "…Those genes are becoming fewer and fewer. So, I feel I have a duty to try to pass them on if possible." (IP18). So, unlike the other men, this man expressed a sense of pride and duty to society to pass on his red-haired, minority genes.

### Rejection – I did not see that coming!

The analysis identified three clusters of responses to rejection, represented in three subthemes: "Significant concerns about new information", "Disappointment over the loss of donor identity", and "Relief of not becoming a donor". In general, the men were satisfied with the information they received both during the application process and in their subsequent rejection.

The men had different responses to the rejection. For five men, the rejection was based on information that gave rise to immediate and significant concerns, e.g., regarding their fertility, their own health, or their children's health. A man in his twenties, who was rejected due to reduced semen quality, said: "I actually felt a little bit, well, almost stunned. I couldn't help but worry about the opportunities that were being closed off for me. I mean, in terms of whether I can have my own children in the future." (IP14). Another man, a man in his thirties and father of two children, rejected due to a genetic predisposition to disease, explained: "I became worried and a bit upset about what the hell was going on with my children now. Whether they were sick or going to develop something bad… or if I was sick myself, and what the hell it actually was." (IP9). For these men, the information produced during the recruitment process produced new insights that introduced significant concerns regarding themselves and their (future) families. These significant concerns overshadowed the rejection itself as a sperm donor.

For the remaining 14 men, the information on which the rejection was based did not give rise to significant health-related concerns. Rather, the analysis showed how it was the rejection itself that made the most impact on the men.

Ten men described reacting with great disappointment. A man in his twenties with no partner or children, said: "I was, of course, disappointed because I had really hoped for it. I was looking forward to it." (IP3). Another younger man explained: "It was kind of a bummer feeling. Mostly because I couldn't become a donor, as I had already started seeing myself as one." (IP10). The analysis showed how some men had already formed an identity around being a sperm donor and were disappointed to have that identity 'denied' or 'taken away'.

Four men expressed disbelief over being – unexpectedly – rejected. A man in his thirties and father of two children explained: "I had kind of just taken it for granted that everything was as it should be, and I could be a donor." (IP2). For others, the disbelief was related to the reason for the rejection, which they felt was based on a 'very minimal' and 'impersonal' foundation. A man in his forties with a partner and no children, said: "A bit of a feeling that my genes were just discarded on paper without further explanation, and without it really making scientific sense." (IP4). He felt that the assessment and subsequent rejection based on a minor physical anomaly was unconvincing and lacked justification.

Six men articulated their disappointment as a sense of defeat and a feeling of 'not being good enough'. A man in his thirties without partner or children, said: "I see it somewhat as a defeat [..]. Ideals I couldn't live up to. What really hurts is the experience that they [accepted donors] have something that I don't." (IP17). These men clearly felt defeat by not 'meeting the standard' of those who become donors. To some, the defeat was linked to their sense of masculinity. A man in his twenties with a partner and no children, said: "It's kind of a blow to masculinity, you could say." (IP12). What these men had in common was that the rejection dealt a blow to their self-esteem and made them feel less worthy than the men who were approved.

Finally, one man, single and in his twenties, described a sense of relief upon being rejected. During the application process, he felt increasingly treated as a commodity: 'It was actually a bit of a relief when I found out. As a donor, I felt like I was just a bunch of data. It fluctuated between being a human and data. A very personal experience is made very impersonal. It's hard to relate to that amount of data which you don't really have any control over. And those who might use it in the future just get a set of papers about who I am." (IP18). This man, highlighting how he felt depersonalized during the

application process and reduced to just data rather than a person, ultimately expressed a preference for not becoming a sperm donor, as the only one in the group.

**Life goes on – it is what it is**

For 14 men, the rejection only affected them for a few days or weeks. By the time of the interview (5–12 months later), it was described as not playing a significant role in their lives. Of the five men for whom the new information had caused significant concerns for themselves and their (future) family, it still 'popped up' in certain situations, such as when discussing family expansion with their partner. When coming to terms with the rejection and the new information that came with it, the men described different strategies that are presented in five clusters below, represented in five subthemes: "Information seeking", "Actionable Reasoning", "Positive Reframing", "Normalizing", and "Postponing".

Two men, both fathers and both rejected due to a genetic predisposition to disease, had been concerned about their own health and the well-being of their children. Following the rejection, they had actively sought additional information about the predispositions, either on the internet or through their personal network: "I have friends in medicine, so I called them as soon as the nurse hung up. I needed to get control of the situation.". (IP9). By reaching out for more detailed and family-specific information – than the information they received through the sperm bank – they were actively trying to reduce uncertainty and the threat posed by this new knowledge. Upon learning that being a carrier posed little risk to their children, they described 'letting go' of their concerns and effectively leaving the rejection behind.

Three men engaged in types of 'actionable reasoning' and attributed the reason for rejection to factors that were within their perceived control, such as their lifestyle choices. A man in his thirties, rejected due to reduced semen quality, said: "I don't know if it's smoking that has affected it or bad eating habits, if I've been drinking a little too much... And those are all factors that I think might have played a role. Maybe I should have waited an extra day before I went in to donate again." (IP7). These men were convinced that the rejection stemmed from their current lifestyle, which they could change if needed. Thus, the rejection was based on a modifiable factor that they perceived as being within their control and this understanding served as a strategy to dismiss or disregard the cause for rejection and its potential consequences.

Eight men used positive reframing as a strategy for coming to terms with the disappointment of the rejection by focusing on the positive aspects of their lives. A man in his forties with a partner and no children, said: "I live in a big, beautiful newly built house, and I would say that I have my life together and on track. I would claim that I am quite a well-functioning, sensible person. In many other respects, I actually think I'm quite attractive, if you can say it that way." (IP4). By shifting his attention away from the rejection and instead highlighting what was going well in his life, he reduced the emotional intensity of the rejection. Some men also focused on the valuable information gained during the application process, e.g., good sperm quality. A man in his twenties, rejected due to a genetic predisposition to disease, said: "I was told that I had super-super good sperm quality, so it was nice to hear that. An ego boost. At least there was something that worked for me." (IP16). This positive information counterbalanced the rejection with a sense of accomplishment and value.

Seven men used normalizing strategies when dealing with the rejection, focusing on other applicants who also did not 'make it'. A man in his twenties, who had been rejected due to reduced semen quality, explained: "Actually, one of my friends received the exact same message, so it was really nice for me not to feel alone in it." (IP10). Other men normalized the rejection by referring to statistics and the majority of applicants that are rejected as sperm donors. A younger man, who had been rejected due to physical illness, said: "Later I found that only about 10% are approved. I didn't know that beforehand. And it was actually really nice to know that you are not the only one. That it's not that strange to be rejected." (IP11). This strategy reduced the feelings of disappointment and 'failure' by shifting focus from individual rejection to a common and shared experience among the majority.

Two men deliberately chose to postpone potential concerns about the cause for rejection, as they did not perceive it as relevant to their current life situation. A man in his twenties, rejected due to reduced semen quality, stated: "For us, children are at least five years in the future. So, I don't feel there's any reason to start a major stress process now. If we

can just do it in four years and then stick to that" (IP14). He did not see any reason to worry or seek follow-up examinations yet. Similarly, another man in his twenties, rejected due to physical illness, said: "I've kind of set it aside for now. I'll reconsider it and maybe see a doctor when having children becomes more relevant." (IP18). This strategy of postponement allowed the young men to reframe the new information as a potential future problem rather than a present problem requiring their attention. As such the new information produced by the rejection was put in transit until circumstances might make it relevant again.

## Discussion

The present study investigated men's experience of being rejected as a sperm donor, focusing on how they cope with the rejection and its cause and the subsequent impact on their lives. We found that the information causing the rejection in some cases raised significant concerns, particularly about health, future fertility, and family. However the analysis also revealed that for many men, it was not the reason for the rejection but the rejection itself that had the greatest impact. They had already identified as a donor, and the rejection thus disrupted their sense of self. Over time, the men used various strategies to integrate the rejection and potential new information into their lives. Ultimately, although the rejection was an emotional setback for many, it did not have a lasting negative impact on their lives.

### Being denied the sperm donor identity

The personal and emotional impact of rejection can be understood from the perspective of Social Identity Theory, developed by Tajfel and Turner (2004). According to the theory, individuals derive part of their identity from their membership in social groups, which in turn influences their behavior and self-esteem. The theory focuses on three key processes: social categorization (classifying oneself and others into groups), social identification (adopting in-group behaviors and norms), and social comparison (comparing in-groups to out-groups, e.g., to boost self-esteem) [21].

In addition to the explicitly stated donor motivations described in theme 1, our analysis in theme 2 revealed how the rejection evoked feelings related to the perceived loss of the donor identity. This suggests that, beyond the altruistic and genetic motivations, many applicants were also driven by the wish to adopt the social identity of a sperm donor. Becoming a sperm donor can be seen as an in-group identity associated with health, virility, and genetic desirability. Previous research has indicated that sperm donation allows men to express responsibility, altruism, and control – qualities that can be associated with masculinity [8,11]. Furthermore, previous research has indicated that discussions surrounding declining sperm counts and male fertility can shape how sperm donors perceive their role, often linking it to notions of virility and genetic quality [10]. However, when rejected, these men experienced a disruption in their self-perception and self-categorization. The rejection denied them their assumed place within the sperm donor community. The disappointment of the rejection can thus be understood as a response to the exclusion from the in-group of accepted donors and an involuntary shift (back) into an out-group. According to Social Identity Theory, individuals who fail to gain acceptance in a desired group often experience a threat to their social identity, which can manifest in emotional distress or frustration [21]. Several participants expressed disappointment over the loss of donor identity, particularly when they found the reason for rejection insufficient or implausible. However, by questioning or challenging the legitimacy of the rejection, the men could maintain a self-identity as 'valuable' and 'attractive' despite the rejection. Furthermore, according to Social Identity Theory, when faced with rejection from a desired group, individuals may attempt to redefine their identity in alternative ways [21]. Our findings similarly showed how the rejection caused some men to recognize and emphasize what they perceived as positive aspects of themselves and their lives and thereby maintaining a self-identity as valuable.

### Managing rejection and its cause

To better understand how the men coped with the concern and/or disappointment of rejection, we looked to the Stress and Coping Model developed by Folkman and Lazarus (1988). This model posits that stress arises from an individual's

appraisal of a situation, which in turn shapes their coping response. When an event, such as sperm donor rejection, is perceived as threatening or challenging, it triggers a stress response. Individuals then employ coping strategies that fall into two broad categories: problem-focused coping, which seeks to address or resolve the issue, and emotion-focused coping, which aims to regulate emotional distress [22].

When viewing the present analysis through the lens of the Stress and Coping Model, it stands out that many men did not engage in problem-focused coping as they simply did not accept or agree with the reason for rejection as a 'problem'. They questioned the legitimacy of the rejection and thus felt no need to take action to solve the 'problem'. They engaged in emotion-focused coping strategies, including positive reframing, normalizing, and postponing to reframe the rejection, positioning it as a neutral life event rather than a personal failure. This allowed them to relatively easily rise above the rejection, reduce distress and move on. This emotion-focused strategy is in line with current research, which found that when people do not perceive new information as a threat, they are more likely to manage their feelings by thinking about the situation in a way that reduces distress [12,24].

For some men in our study, the rejection came with new health or genetic information that raised significant concerns about their own or their children's well-being. These men engaged in problem-focused coping strategies, such as seeking information and engaging in actionable reasoning. Information-seeking behavior, in particular, was a prominent response, as men attempted to reduce uncertainty and regain control through information and knowledge. This aligns with research indicating that when unexpected health information is received, individuals often engage in proactive information-seeking to reframe the situation and reduce perceived risk [7,11]. Moreover, problem-focused coping has been widely associated with better psychological adjustment in situations where control is attainable [25], which may explain why some men were able to reappraise the rejection in a way that reduced its emotional impact. Based on these findings, sperm banks could consider interventions to support men who are rejected, such as optional appointments with a psychologist, group support sessions, or digital tools for guidance and emotional support. While further research is needed to evaluate the effectiveness of such measures, they could help ensure all men feel supported and well-informed throughout the donation process. Since such new, medical information can cause significant concern among rejected donors, sperm banks should support affected men and ensure that they leave the process positively.

## Strength and limitations

A strength of this study is the explorative, qualitative approach whereby the participating men were encouraged to share their individual reactions and coping strategies, which allowed for an in-depth understanding not restricted by predefined options of a questionnaire. By applying Social Identity Theory and the Stress and Coping model, we were able to uncover the psychological mechanisms behind the men's reactions, providing deeper insight into the emotional consequences of donor rejection. One limitation is that the present study is relatively small. However, in qualitative research there is no uniform standard for assessment of adequate sample size, but the concept of "information power" advocates that the more information a sample holds relevant for the study, the lower the number of participants needed [26]. Another limitation is that all participants voluntarily agreed to take part in the study. This means that the findings may primarily reflect the perspectives of men who felt particularly affected by the rejection or had a strong desire to share their experiences. Those who were less impacted or chose not to engage may have different reactions and coping strategies that were not captured in this study. Lastly, qualitative studies are not generalizable in the quantitative sense of the term and should always be interpreted within their specific context. By clearly defining the study population and making the context explicit and transparent, we aim to enable readers to meaningfully assess the relevance and applicability of our findings to their own settings.

## Conclusion

The information prompting rejection may raise medical concerns in donor candidates. However, over time, the men engaged in various strategies to process the experience, and although the rejection was initially distressing, it did not

result in lasting negative consequences. Beyond personal impact, future research could explore how newly acquired health or genetic information affects not only the men but also their families. This information could influence their relationships, family planning, and perceptions of their roles as fathers or partners, shedding light on the broader emotional and psychological consequences of donor rejection.

## Acknowledgments

The authors wish to thank all the participating men who took time to share their experiences with us.

## Author contributions

**Conceptualization:** Lina Thirup, Anne-Bine Skytte, Ulrik Schiøler Kesmodel, Ida Vogel, Guido Pennings, Stina Lou.

**Data curation:** Lina Thirup, Anne-Bine Skytte, Stina Lou.

**Formal analysis:** Lina Thirup, Stina Lou.

**Funding acquisition:** Anne-Bine Skytte.

**Project administration:** Lina Thirup, Stina Lou.

**Supervision:** Anne-Bine Skytte, Ulrik Schiøler Kesmodel, Ida Vogel, Guido Pennings, Allan Pacey.

**Validation:** Anne-Bine Skytte, Ulrik Schiøler Kesmodel, Ida Vogel, Guido Pennings, Allan Pacey.

**Writing – original draft:** Lina Thirup, Stina Lou.

**Writing – review & editing:** Anne-Bine Skytte, Ulrik Schiøler Kesmodel, Ida Vogel, Guido Pennings, Allan Pacey.

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
