## [Decision Letter · Decision Letter 0]

3 Sep 2025

Dear Dr. Thirup,

**The paper has interesting aspects but suffers from several shortcomings. The authors should try to consider all aspects raised by the reviewers during revision.**

We look forward to receiving your revised manuscript.

Kind regards,

Stefan Schlatt

Academic Editor

PLOS ONE

**Journal Requirements:**

1. When submitting your revision, we need you to address these additional requirements. Please ensure that your manuscript meets PLOS ONE's style requirements, including those for file naming. The PLOS ONE style templates can be found at https://journals.plos.org/plosone/s/file?id=wjVg/PLOSOne_formatting_sample_main_body.pdf and https://journals.plos.org/plosone/s/file?id=ba62/PLOSOne_formatting_sample_title_authors_affiliations.pdf 2. Thank you for stating the following in the Competing Interests section: I have read the journal's policy and the authors of this manuscript have the following competing interests: LT is a former employee of Cryos International Sperm & Egg bank. ABS and USK are employed by Cryos International Sperm & Egg bank. GP and AP are members of the External Scientific Advisory Committee (ESAC) of Cryos. IV and SL have declared that no competing interests exist.  We note that one or more of the authors are employed by a commercial company.  a. Please provide an amended Funding Statement declaring this commercial affiliation, as well as a statement regarding the Role of Funders in your study. If the funding organization did not play a role in the study design, data collection and analysis, decision to publish, or preparation of the manuscript and only provided financial support in the form of authors' salaries and/or research materials, please review your statements relating to the author contributions, and ensure you have specifically and accurately indicated the role(s) that these authors had in your study. You can update author roles in the Author Contributions section of the online submission form. Please also include the following statement within your amended Funding Statement. “The funder provided support in the form of salaries for authors, but did not have any additional role in the study design, data collection and analysis, decision to publish, or preparation of the manuscript. The specific roles of these authors are articulated in the ‘author contributions’ section.”If your commercial affiliation did play a role in your study, please state and explain this role within your updated Funding Statement.  b. Please also provide an updated Competing Interests Statement declaring this commercial affiliation along with any other relevant declarations relating to employment, consultancy, patents, products in development, or marketed products, etc.   Within your Competing Interests Statement, please confirm that this commercial affiliation does not alter your adherence to all PLOS ONE policies on sharing data and materials by including the following statement: "This does not alter our adherence to  PLOS ONE policies on sharing data and materials.” (as detailed online in our guide for authors http://journals.plos.org/plosone/s/competing-interests) . If this adherence statement is not accurate and  there are restrictions on sharing of data and/or materials, please state these. Please note that we cannot proceed with consideration of your article until this information has been declared. Please include both an updated Funding Statement and Competing Interests Statement in your cover letter. We will change the online submission form on your behalf. 3. If the reviewer comments include a recommendation to cite specific previously published works, please review and evaluate these publications to determine whether they are relevant and should be cited. There is no requirement to cite these works unless the editor has indicated otherwise. 

**Additional Editor Comments:**

The paper looks at an interesting aspect of sperm donors. However the referees raise doubts towards the validity of the approach. The authors should revise their manuscript and try to improve validity by better data handling and analysis.

Reviewers' comments:

**Comments to the Author**

1. Is the manuscript technically sound, and do the data support the conclusions?

Reviewer #1: Yes

Reviewer #2: Partly

2. Has the statistical analysis been performed appropriately and rigorously?

Reviewer #1: N/A

Reviewer #2: N/A

3. Have the authors made all data underlying the findings in their manuscript fully available?

Reviewer #1: Yes

Reviewer #2: No

4. Is the manuscript presented in an intelligible fashion and written in standard English?

Reviewer #1: Yes

Reviewer #2: Yes

**Reviewer #1:**  Dear authors,

Congratulations to this interesting manuscript about a very important aspect of sperm donations. I have three minor revision points, which I think would make the manuscript even stronger.

As a clinician I would ask for a personal outlook regarding this data set. No we the authors know how the men felt, but what could be changed in the process that all of them feel well after being rejected (appointments with psychologist, group therapy, digitial apps, ...)? I would adivse the authors to include an outlook in the last parts of the article.

It would be interesting to know how many sperm donations were recieved in total in the mentioned time frame. Were the 188 rejected ones taken from 1000 or 2000 sperm donations.

Lastly, it would be thrilling to know anything about the sperm quality and the underlying diseases. In my personal view it does make a difference if azoospermia is found or if OAT is found. Furthermore, it does make a difference if for expample cystic fibrosis is found or a pre-disposition to type 1 diabetes. I would encourage the authors to include more date on these two topics.

**Reviewer #2:**  1. As written in the limitations by the authors, it is a small and qualitative study. I believe the results shown are valuable in their explorative nature, but the data cannot be used in a quantitative way. Certain data analysis tools (thematic analysis and “Nvivo 15 software”) are used to cluster the results, but I am not sufficiently familiar with these tools to judge how appropriate this use is.

2. No statistical analysis was done

3. As mentioned in the manuscipt, for privacy reasons the transcripts could not be shared. Instead, the themes are analysed and the number of participants who fit into each theme are described in the text.

4. The writing style is good and easy to understand.

Additional comments:

Table 1 has Age (years), mean (range); time since rejection (months), mean (range) and duration of interview (minutes), mean (range) as descriptors. However, these should be given as Mean age in years (range); mean time since rejeaction in months (range), mean duration of interview in minutes (range) or other less confusing formatting.

In the introduction, only the infertility crisis is given as a reason for the increase in sperm donor demand, even though studies show most of the demand for sperm donors today is from same-sex couples and solo mothers. The reference given (1) is also specifically about sperm quality decreasing, which is not the sole cause of the “infertility crisis”. Overall, the introduction could benefit from a broader selection of references.

Table 1 and Table 3 could be combined: The characteristics and reasons for rejection of each donor were likely known by the authors before the survey was done, so are not really "results" as such, unless they are the self-reported reasons for rejection. If they were indeed self-reported (and as such part of the results), the text could go into more detail about whether or not the self-reported reasons matched those on file by the sperm bank.

Except for Table 3, the results are described in text form split into the three themes, with a schematic of the themes shown in Figure 1. Figure 1 could be expanded (or an additional figure created) to give an overview of the 19 donors, perhaps with a colour code, and which specific donor identified with which theme.

Throughout the manuscript, the specific authors who conducted specific parts of the study are mentioned. It would be better if the methods were described objectively and the author contributions at the end listed the specific roles of each author.

Overall, I think these types of surveys and their analysis provide very valuable insights into the social and ethical implications of gamete donation (even for those who do not become donors, as is the case here), but the manuscript would benefit from some major revisions to elevate the quality of how results are presented.

**Do you want your identity to be public for this peer review?** For information about this choice, including consent withdrawal, please see our Privacy Policy

Reviewer #1: No

Reviewer #2: **Yes: ** Julia Uraji

---

## [Author Response · Author response to Decision Letter 1]

2 Oct 2025

Dear Stefan Schlatt, Academic Editor, PLOS ONE

We would like to sincerely thank you and the reviewers for the insightful and constructive feedback on our manuscript. We have carefully revised the paper in line with the reviewers’ suggestions, which we believe has strengthened the manuscript considerably.

Below, we provide a detailed response to each comment and outline how we have addressed them in the revised version of the manuscript. All line numbers refer to the tracked-changes version.

We hope that the revisions meet your expectations and we look forward to your response.

Best regards,

Lina Thirup

Journal Requirements

Please ensure that your manuscript meets PLOS ONE’s style requirements, including those for file naming. The PLOS ONE style templates can be found at

We have revised the manuscript in accordance with PLOS ONE’s formatting requirements, including those for file naming.

I have read the journal’s policy and the authors of this manuscript have the following competing interests: LT is a former employee of Cryos International Sperm & Egg bank. ABS and USK are employed by Cryos International Sperm & Egg bank. GP and AP are members of the External Scientific Advisory Committee (ESAC) of Cryos. IV and SL have declared that no competing interests exist.

We note that one or more of the authors are employed by a commercial company.

a. Please provide an amended Funding Statement declaring this commercial affiliation, as well as a statement regarding the Role of Funders in your study. If the funding organization did not play a role in the study design, data collection and analysis, decision to publish, or preparation of the manuscript and only provided financial support in the form of authors’ salaries and/or research materials, please review your statements relating to the author contributions, and ensure you have specifically and accurately indicated the role(s) that these authors had in your study. You can update author roles in the Author Contributions section of the online submission form.

Please also include the following statement within your amended Funding Statement. “The funder provided support in the form of salaries for authors but did not have any additional role in the study design, data collection and analysis, decision to publish, or preparation of the manuscript. The specific roles of these authors are articulated in the ‘author contributions’ section.”

a.

Updated Funding Statement:

This study was conducted as part of the authors’ regular employment at Cryos International Sperm & Egg Bank, a commercial company. No specific project funding was provided beyond the authors’ salaries. The authors (LT, ABS and USK) are employed by Cryos International Sperm & Egg Bank and receive salaries for a range of tasks unrelated to this specific research project. The management of Cryos International supported the idea for the project but did not have any additional role in the study design, data collection and analysis, decision to publish, or preparation of the manuscript. The specific roles of these authors are described in the ‘Author Contributions’ section.

Updated Author Contributions:

Conceptualization: Lina Thirup, Anne-Bine Skytte, Ulrik Schiøler Kesmodel, Ida Vogel, Guido Pennings, Allan Pacey, Stina Lou.

Data curation: Lina Thirup, Anne-Bine Skytte, Stina Lou.

Formal analysis: Lina Thirup, Stina Lou.

Funding acquisition: Anne-Bine Skytte.

Project administration: Lina Thirup, Stina Lou.

Supervision: Anne-Bine Skytte, Ulrik Schiøler Kesmodel, Ida Vogel, Guido Pennings, Allan Pacey.

Validation: Anne-Bine Skytte, Ulrik Schiøler Kesmodel, Ida Vogel, Guido Pennings, Allan Pacey.

Writing – original draft: Lina Thirup, Stina Lou.

Writing – review & editing: Anne-Bine Skytte, Ulrik Schiøler Kesmodel, Ida Vogel, Guido Pennings, Allan Pacey.

The funder (Cryos International Sperm & Egg Bank) provided support in the form of salaries for some authors but had no additional role in study design, data collection and analysis, decision to publish, or preparation of the manuscript.

b. Please also provide an updated Competing Interests Statement declaring this commercial affiliation along with any other relevant declarations relation to employment, consultancy, patents, products in development, or marketed products, etc.

Within your Competing Interests Statement, please confirm that this commercial affiliation does not alter your adherence to all PLOS ONE policies on sharing data and materials by including the following statement: “This does not alter our adherence to PLOS ONE policies on sharing data and materias.” (as detailed online in our guide for authors http://journals.plos.org/plosone/s/competing-interests). If this adherence statement is not accurate and there are restrictions on sharing of data and/or materials, please state these. Please note that we cannot proceed with consideration of your article until this information has been declared.

b.

Updated Competing Interests:

I have read the journal's policy and the authors of this manuscript have the following competing interests: LT is a former employee of Cryos International Sperm & Egg Bank. ABS and USK are employed by Cryos International Sperm & Egg Bank. GP and AP are members of the External Scientific Advisory Committee (ESAC) of Cryos. IV and SL have declared that no competing interests exist. This does not alter our adherence to PLOS ONE policies on sharing data and materials. Due to privacy protection, data material can only be made available upon reasonable request to Cryos International Sperm & Egg Bank.

The reviewers did not suggest specific references or previous work to cite. However, in response to their comments, we have added several relevant references to broaden the introduction and reflect the growing demand for donor sperm among single women and same-sex couples (lines 63–66).

Additional Editor Comments

The paper looks at an interesting aspect of sperm donors. However the referees raise doubts towards the validity of the approach. The authors should revise their manuscript and try to improve validity by better data handling and analysis.

We thank the reviewer for highlighting this point. In the revised manuscript, we have clarified our data handling and analysis procedures, emphasized the qualitative and exploratory nature of the study, and explained that the findings are not intended for quantitative generalization (lines 127-129, 406-409).

Comments to the Author

1. Is the manuscript technically sound, and do the data support the conclusions?

Reviewer #1: Yes

Reviewer #2: Partly

Please see response to reviewer #2 below.

2. Has the statistical analysis been performed appropriately and rigorously?

Reviewer #1: N/A

Reviewer #2: N/A

Not applicable, as this is a qualitative study and no statistical analyses were performed.

3. Have the authors made all data underlying the findings in their manuscript fully available?

Reviewer #1: Yes

Reviewer #2: No

Please see response to reviewer #2 below.

4. Is the manuscript presented in an intelligible fashion and written in standard English?

Reviewer #1: Yes

Reviewer #2: Yes

We are pleased that the reviewers have confirmed that the manuscript is clearly written in standard English.

5. Review Comments to the Author

Reviewer #1

Congratulations to this interesting manuscript about a very important aspect of sperm donations. I have three minor revision points, which I think would make the manuscript even stronger.

We appreciate the reviewer’s positive feedback and constructive suggestions. All points have been addressed in the revised manuscript.

As a clinician I would ask for a personal outlook regarding this data set. No we the authors know how the men felt, but what could be changed in the process that all of them feel well after being rejected (appointments with psychologist, group therapy, digitial apps, ...)? I would adivse the authors to include an outlook in the last parts of the article.

We thank the reviewer for this helpful suggestion. In the revised manuscript (lines 384–388), we have added a brief outlook discussing potential interventions to support men who are rejected, such as optional appointments with a psychologist, group support sessions, or digital tools. These measures are suggested to help ensure that all men feel supported and well-informed throughout the donation process.

It would be interesting to know how many sperm donations were recieved in total in the mentioned time frame. Were the 188 rejected ones taken from 1000 or 2000 sperm donations. We appreciate the reviewer’s suggestion.

We have now clarified in the Methods section that the 188 contacted rejected donors were drawn from a total of 3,031 men who applied to become sperm donors during the one-year period from 26 April 2023 to 26 April 2024 (lines 96–98).

Lastly, it would be thrilling to know anything about the sperm quality and the underlying diseases. In my personal view it does make a difference if azoospermia is found or if OAT is found. Furthermore, it does make a difference if for expample cystic fibrosis is found or a pre-disposition to type 1 diabetes. I would encourage the authors to include more date on these two topics.

We agree. However, due to the small sample size (n = 19) and the highly individual and sensitive nature of the information, providing more detailed data on sperm quality and underlying diseases would risk compromising participant anonymity. We have therefore chosen to present these data only in aggregated categories. Nevertheless, where it is relevant for the analysis, we have sought to provide limited, non-identifiable information about the participants to give context while still safeguarding confidentiality.

Reviewer #2

1. As written in the limitations by the authors, it is a small and qualitative study. I believe the results shown are valuable in their explorative nature, but the data cannot be used in a quantitative way. Certain data analysis tools (thematic analysis and “Nvivo 15 software”) are used to cluster the results, but I am not sufficiently familiar with these tools to judge how appropriate this use is.

We thank the reviewer for acknowledging the exploratory value of our study. We have clarified in the revised manuscript (lines 127–129) that NVivo 15 was used solely to organize and manage the qualitative data, and that all coding decisions were reviewed and discussed among the authors to ensure rigor. We also emphasize (lines 406-409) that the study is qualitative and explorative, and the results are not intended for quantitative generalization. We hope these clarifications address your concerns regarding the validity of our approach.

2. No statistical analysis was done

Not applicable, as this is a qualitative study and no statistical analyses were performed.

3. As mentioned in the manuscipt, for privacy reasons the transcripts could not be shared. Instead, the themes are analysed and the number of participants who fit into each theme are described in the text.

We thank the reviewer for acknowledging the importance of data availability. As described in the manuscript, full transcripts cannot be shared publicly due to participant confidentiality. Instead, we have provided a detailed analysis of themes and the number of participants corresponding to each theme in the text. Anonymized data may be made available upon reasonable request from Cryos International (contact: dk@cryosinternational.com) for researchers who meet the criteria for access to confidential data, as outlined in our Data Availability Statement.

4. The writing style is good and easy to understand

We are pleased that the reviewers have confirmed that the manuscript is clearly written in standard English.

Table 1 has Age (years), mean (range); time since rejection (months), mean (range) and duration of interview (minutes), mean (range) as descriptors. However, these should be given as Mean age in years (range); mean time since rejeaction in months (range), mean duration of interview in minutes (range) or other less confusing formatting.

Good point! Table 1 has been revised to improve clarity, and the descriptors are now formatted as suggested: Mean age in years (range); mean time since rejection in months (range); mean duration of interview in minutes (range).

In the introduction, only the infertility crisis is given as a reason for the increase in sperm donor demand, even though studies show most of the demand for sperm donors today is from same-sex couples and solo mothers. The reference given (1) is also specifically about sperm quality decreasing, which is not the sole cause of the “infertility crisis”. Overall, the introduction could benefit from a broader selection of references.

Very important point! We have revised the introduction (lines 63–66) to acknowledge that a substantial proportion of the demand for donor sperm today comes from single women and same-sex couples, and we have added relevant references to reflect this broader perspective.

Table 1 and Table 3 could be combined: The characteristics and reasons for rejection of each donor were likely known by the authors before the survey was done, so are not really "results" as such, unless they are the self-reported reasons for rejection. If they were indeed self-reported (and as such part of the results), the text could go into more detail about whether or not the self-reported reasons matched those on file by the sperm bank.

We thank the reviewer for this helpful suggestion. We have now incorporated the information from Table 3 into Table 1 to present the data more clearly.

Except for Table 3, the results are described in text form split into the three themes, with a schematic of the themes shown in Figure 1. Figure 1 could be expanded (or an additional figure created) to give an overview of the 19 donors, perhaps with a colour code, and which specific donor identified with which theme.

We have carefully considered the reviewer’s suggestion and thank them for t

---

## [Decision Letter · Decision Letter 1]

24 Nov 2025

Coping with rejection as a sperm donor: A qualitative study of the personal impact of rejection and new health information

PONE-D-25-31453R1

Dear Dr. Thirup,

We’re pleased to inform you that your manuscript has been judged scientifically suitable for publication and will be formally accepted for publication once it meets all outstanding technical requirements.

Kind regards,

Stefan Schlatt

Academic Editor

PLOS ONE

Additional Editor Comments (optional):

Reviewers' comments:

Reviewer's Responses to Questions

**Comments to the Author**

Reviewer #1: All comments have been addressed

2. Is the manuscript technically sound, and do the data support the conclusions?

Reviewer #1: Yes

3. Has the statistical analysis been performed appropriately and rigorously?

Reviewer #1: Yes

4. Have the authors made all data underlying the findings in their manuscript fully available?

Reviewer #1: Yes

5. Is the manuscript presented in an intelligible fashion and written in standard English?

Reviewer #1: Yes

Reviewer #1: The manuscript improved. Congratulations to the authors on this valuable manuscript! It will benefit the field of reproductive health.

**Do you want your identity to be public for this peer review?** For information about this choice, including consent withdrawal, please see our Privacy Policy

Reviewer #1: No

---

## [Editor Report · Acceptance letter]

PONE-D-25-31453R1

PLOS ONE

Dear Dr. Thirup,

I'm pleased to inform you that your manuscript has been deemed suitable for publication in PLOS ONE. Congratulations! Your manuscript is now being handed over to our production team.

Kind regards,

on behalf of

Dr. Stefan Schlatt

Academic Editor

PLOS ONE